Analysis and comparison of protein secondary structures in the rachis of avian flight feathers

Lin Pin-Yen 1
Huang Pei-Yu 2
Lee Yao-Chang yclee@nsrrc.org.tw 2 3
Ng Chen Siang gcsng@life.nthu.edu.tw 1 4 5 6
1 Institute of Molecular and Cellular Biology, National Tsing Hua University , Hsinchu , Taiwan
2 National Synchrotron Radiation Research Center , Hsinchu , Taiwan
3 Department of Optics and Photonics, National Central University , Chung-Li , Taoyuan , Taiwan
4 Department of Life Science, National Tsing Hua University , Hsinchu , Taiwan
5 Bioresource Conservation Research Center, National Tsing Hua University , Hsinchu , Taiwan
6 The iEGG and Animal Biotechnology Center, National Chung Hsing University , Taichung , Taiwan
Uversky Vladimir
Electronic publication date: 2022 Feb 28
Publication date: 2022
Volume: 10
Electronic Location ID: e12919
Received 2021 Oct 29; Accepted 2022 Jan 20
Copyright: ©2022 Lin et al.
Copyright year: 2022
Copyright holder: Lin et al.
License: This is an open access article distributed under the terms of the Creative Commons Attribution License, which permits unrestricted use, distribution, reproduction and adaptation in any medium and for any purpose provided that it is properly attributed. For attribution, the original author(s), title, publication source (PeerJ) and either DOI or URL of the article must be cited.
License URL: https://creativecommons.org/licenses/by/4.0/

Keywords: Protein, Secondary structure, Birds, Feather, Rachis, Flight

Funding: Ministry of Science and Technology, Taiwan MOST 107-2311-B-007-008-MY3 MOST 110-2628-B-007-001 iEGG and Animal Biotechnology Center of National Chung Hsing University This work was supported by the Ministry of Science and Technology, Taiwan (Grant Nos. MOST 107-2311-B-007-008-MY3 & MOST 110-2628-B-007-001 to C.S.N.); and the iEGG and Animal Biotechnology Center of National Chung Hsing University and Bioresource Conservation Research Center of National Tsing Hua University from the Feature Areas Research Center Program within the framework of the Higher Education Sprout Project by the Ministry of Education, Taiwan. The funders had no role in study design, data collection and analysis, decision to publish, or preparation of the manuscript.

==============================
Avians have evolved many different modes of flying as well as various types of feathers for adapting to varied environments. However, the protein content and ratio of protein secondary structures (PSSs) in mature flight feathers are less understood. Further research is needed to understand the proportions of PSSs in feather shafts adapted to various flight modes in different avian species. Flight feathers were analyzed in chicken, mallard, sacred ibis, crested goshawk, collared scops owl, budgie, and zebra finch to investigate the PSSs that have evolved in the feather cortex and medulla by using nondestructive attenuated total reflection Fourier transform infrared spectroscopy (ATR-FTIR). In addition, synchrotron radiation-based, Fourier transform infrared microspectroscopy (SR-FTIRM) was utilized to measure and analyze cross-sections of the feather shafts of seven bird species at a high lateral resolution to resolve the composition of proteins distributed within the sampled area of interest. In this study, significant amounts of α-keratin and collagen components were observed in flight feather shafts, suggesting that these proteins play significant roles in the mechanical strength of flight feathers. This investigation increases our understanding of adaptations to flight by elucidating the structural and mechanistic basis of the feather composition.

Introduction

Feathers are rigid, complex structures in the epidermis of dinosaurs and birds (Chen et al., 2015; Ng & Li, 2018). The earliest evidence of feathers has been suggested to derive from paleontological investigations of the fossils of Late Jurassic Theropod dinosaurs (Xu et al., 2014). The feathers collected from different species of birds have many distinctions in shape, size, internal structure, and color; therefore, observing and comparing various feathers helps to further understand the characteristics of feathers among bird species and their functions.

The original function of feathers has been suggested not to be flight, as nonavian dinosaurs lacked the physical adaptations necessary for flying (Chuong et al., 2003). The mechanics of bird flight and energy consumption for this purpose are highly related to the intrastructure and morphology of feathers (Sullivan, Meyers & Arzt, 2019). To satisfy the aerodynamic requirements of the function of avian flight, the component material and mechanical properties of flight feathers are critical (Lingham-Soliar, 2014; Sullivan et al., 2016; Sullivan et al., 2017).

The flight feathers of avians consist of a central shaft and vanes on both sides (Prum & Brush, 2002; Chen et al., 2015), and the size of the inner vane is greater than that of the outer vane to reduce the resistance from aerodynamic forces (Feo, Field & Prum, 2015). The vanes are composed of barbs in a parallel arrangement, and numerous barbules are found on both sides of the barbs. The bow barbules have a dorsal spine, and the hook barbules have many hooklets, allowing adjacent barbs to be interlocked to form a complete vane (Zhang, Jiang & Wang, 2018), which can be separated into two parts by pulling the barbs apart and restored by gently stroking the feather (Kovalev, Filippov & Gorb, 2014; Zhang, Jiang & Wang, 2018).

The shaft of flying feathers is divided into two parts: the rachis and calamus (Fig. 1). Tough and lightweight feathers are composed of keratins and corneous proteins, and the fabric structure and multilayered arrangement of the cortex in the rachis confer it with good elasticity and resistance to fractures (Lingham-Soliar, Bonser & Wesley-Smith, 2010; Laurent et al., 2014; Lingham-Soliar, 2017). Although previous studies showed that the flexural stiffness of feathers could be mainly determined by the structure of the rachises rather than the material properties of keratins and corneous proteins (Bonser & Purslow, 1995; Weiss, Schmitt & Kirchner, 2011; Bachmann et al., 2012), the external morphology is not the single major factor determining rachis structural properties, but a combination of microstructural and material properties could be critical (Lees et al., 2017). In addition, the rachis has a less significant effect on the buckling stress and tensile strength of the rachis, which will affect the bending of the feathers to prevent the rachis from being cracked by external forces during flight (Wang & Meyers, 2017; Zou et al., 2019).

Figure 1 The external and internal structure of the feather shaft of a bird.

(A) Feathers of domestic chickens. The shaft of flight feathers can be divided into two parts: rachis and calamus. 0.0 Z to 1.0 Z represents the one-dimensional position of the rachis from the proximal part to the distal part, and 0.4 Z is the position of the slice in the figure. (B) A slice of the rachis (0.4 Z). The rachis has two parts: cortex and medulla.

The laterally resolved distribution of α- and β-keratin in mature flight feathers can provide two-dimensional information for understanding the mechanical requirements of the flight feathers to meet the aerodynamic needs of birds. This study also provides a novel perspective on the adaptation of the rachis medulla for flight. Various strategies and methods related to flight mechanics, such as gliding, hovering, bounding, flocking, and flapping, have been well investigated in different kinds of avians (Rayner, 1982). The distribution pattern of the change in the mechanical force acting upon the wing is influenced by various flight modes. The loading of aerodynamic forces is applied mostly to the flight feather shaft; hence, the structural properties of rachises vary across the wingspan and among different birds (Ennos, Hickson & Roberts, 1995; Pap et al., 2015; Osvath et al., 2020).

The number of corneous proteins invested in flight feathers and their structure varies with flying behavior and life history (Worcester, 1996; Aparicio, Bonal & Cordero, 2003; Lingham-Soliar, Bonser & Wesley-Smith, 2010; Weber et al., 2010; Bachmann et al., 2012; Wang et al., 2012). The mechanism of keratin and corneous protein accumulation in the development of feathers needs to be further understood. The major components of feathers are reported as α- and β-keratin, encoded by multigene families (Shames & Sawyer, 1987; Ng et al., 2014; Alibardi, 2017). However, the components of α-keratins are found in all vertebrates, while those of β-keratins are only observed in birds and reptiles. β-Keratin is unrelated to α-keratin and likely emerged after divergence between the sauropsid and mammalian lineages (Alibardi & Sawyer, 2002; Sawyer et al., 2005; Wu, Irwin & Zhang, 2008; Greenwold & Sawyer, 2010; Fraser & Parry, 2011; Strasser et al., 2014; Holthaus et al., 2018a; Holthaus et al., 2018b). β-Keratins have been proposed to be renamed corneous β-proteins (CBPs) (Alibardi, 2016; Holthaus et al., 2018a; Holthaus et al., 2018b); however, recent publications still conventionally refer to them as β-keratins (Chen et al., 2021; Lin et al., 2021a; Lin et al., 2021b; Saranathan et al., 2021; Zhang & Fan, 2021).

β-keratins are mainly observed as infrastructural material expressed in the cortex, whereas α-keratins are generally expressed in the medulla (Ng et al., 2014). Both typical enhancers and temporospatial chromatin looping precisely regulate the expression of β-keratin gene clusters to establish the macro-regional specification of feathers and micro-level regional specification of feathers, respectively (Liang et al., 2020; Chen et al., 2021). An X-ray diffraction approach showed that the filament has a helical structure with four repeating units per turn, and hairpin turns in the β-sheet were also identified and shown to be unusually rich in proline residues (Fraser & Parry, 2008). Furthermore, theoretical simulations predicted that the length of the β-keratin chain is typically 100–200 molecules and contains a central conserved region of approximately 34 amino acid residues, enabling the adoption of a twisted antiparallel β-sheet conformation with three central strands and two partial outer strands (Fraser & Parry, 2011; Calvaresi, Eckhart & Alibardi, 2016).

Most studies have applied cell biology approaches for exploring the protein composition of feathers or the structures of the polypeptide chain in the protein (Alibardi, 2017), but it is still difficult to reveal the differences in related protein secondary structures (PSSs) among different types of feathers. Several studies have applied microbial biodegradation followed by scanning electron microscopy (SEM) measurements (Lingham-Soliar, Bonser & Wesley-Smith, 2010; Lingham-Soliar, 2014; Lingham-Soliar, 2017), X-ray CT scanning (Laurent et al., 2014; Chang et al., 2019; Laurent et al., 2020a), infrared microspectroscopy (Tsuboi et al., 1991), or Raman spectroscopy (Tsuboi et al., 1991; Laurent et al., 2020b) to obtain morphological and PSS composition information in different positions in the rachises of various birds, but they have mainly focused on analyzing the microstructural architecture or β-pleated sheets of β-keratins.

Differences in the morphology, intrastructure, and composition of the feather shafts of birds have been proposed to correlate with different flight abilities and modes. Herein, we investigated the proportions of keratins and corneous proteins, which can be represented by PSSs and are likely to be affected by flying modes. Fourier transform infrared (FTIR) microspectroscopy is a state-of-the-art technology that can be employed as a nondestructive analytical tool for obtaining the spatially resolved absorption of functional groups of biomolecules distributed within the area of interest in feather samples. Over decades, FTIR technologies have been widely utilized to analyze PSSs (Cai & Singh, 1999; Lopez-Lorente & Mizaikoff, 2016), and the molecular vibration frequencies of the main or side chain of the polypeptide can be resolved to elucidate the relationship between the percentages of PSSs and the flight modes of different species. The characteristic absorption bands of amide I (Am I) and amide II (Am II) are mainly attributed to carbonyl groups (>C=O) and C-N-H bonds of the peptide bonds for establishing PSSs by a hydrogen bonding framework in protein molecules; in particular, the Am I band has been widely used to resolve the different components of PSSs (Yang et al., 2015; Andrew Chan & Kazarian, 2016; Lopez-Lorente & Mizaikoff, 2016; Tucureanu, Matei & Avram, 2016). The absorption profile of the Am I band contains abundant information about PSSs, including α-helix, β-sheet, β-turn, and random coil structures related to the carbonyl groups of peptide bonds in different physical and/or chemical environments in the frequency range of 1,720–1,580 cm−1. Moreover, PSSs in the Am I band can be unfolded by employing Fourier self-deconvolution and second derivatives of the absorption profile of the Am I band to quantitatively analyze the population of each PSS.

In addition to α- and β-keratin, collagen genes have been determined to be expressed in growing feathers (Ng et al., 2015). Collagen type I and type II are known to be involved in the feathering of chicken embryos (Mauger et al., 1982). Among them, collagen type I is the major protein in connective tissues and the most abundant protein in vertebrates. The structure of collagen consists of three-stranded, left-handed procollagen. Glycine forms a right-handed structure in which hydrogen bonds are entangled with each other. This feature of the triple helix can be observed in the infrared absorption spectrum at 1,637 cm−1 (Cooper & Knutson, 1995; Belbachir et al., 2009; Lee et al., 2017).

In this study, synchrotron-radiation-based Fourier transform infrared microspectroscopy (SR-FTIRM) was employed to acquire spatially resolved FTIR spectra and to reveal the biomaterial composition distributed within the feather sections of the rachis cortex and medulla of seven species of birds using the Peak Resolve program in OMNIC™ software (Version 9.2; Thermo Fisher Scientific Inc., Waltham, MA, USA) to elucidate the PSSs. The initial value of the band center for each possible component of PSSs was found by the second derivative of the curve of the FTIR spectrum in the spectral range of 1,720–1,580 cm−1 of the feather sample. The spectral components of PSSs in the Am I band of the FTIR spectra acquired for feather samples from bird species were quantitatively calculated by iterating the curve-fitting process using a Gaussian function profile, and the line width of the full width at half maximum (FWHM) was set to 30 cm−1 for the initial fitting process in the Peak Resolve program. FTIR spectra were baseline-corrected before the process, and the curve fitting was processed by iteration in the calculation loop for adjusting parameter variables to minimize standard errors of the set of parameters, including FWHM, peak height, the peak position of the absorption component peaks, and the peak areas contributed from the corresponding absorption component peaks. Finally, the fitting result was obtained based on the previously mentioned parameters and each peak area of the corresponding components of PSSs.

The seven bird species investigated herein were chicken (Gallus gallus domesticus), mallard duck (Anas platyrhynchos), African sacred ibis (Threskiornis aethiopicus), crested goshawk (Accipiter trivirgatus), collared scops owl (Otus lettia), budgerigar (Melopsittacus undulatus), and zebra finch (Taeniopygia guttata). Using the flight feathers of these birds allowed us to detect the common basis of the proportions of birds with different flight modes.

Materials & Methods

Feather specimens

All the feathers analyzed in this study were primary flight feathers, including domestic chicken (Gallus gallus domesticus), mallard duck (Anas platyrhynchos), African sacred ibis (Threskiornis aethiopicus), crested goshawk (Accipiter trivirgatus), collared scops owl (Otus lettia), budgerigar (Melopsittacus undulatus), and zebra finch (Taeniopygia guttata), to represent bird species with different flight modes. Primary flight feathers of domestic chickens and mallards were collected from an aviary at National Chung Hsing University. Primary flight feathers of budgerigar and zebra finch were collected from an aviary at National Tsing Hua University. Primary flight feathers of African sacred ibis, crested goshawk, and collared scops owl were plucked from the carcasses of wild individuals. The use of feathers from wild animals for research was approved by the Forestry Bureau of the Council of Agriculture of Taiwan (R.O.C.) under case no. 1091617349 (4/22/2020).

Paraffin-embedded feather sections

The sectioned feather samples from a serial 5-µm thick section of the paraffin embedding rachis of flight feathers were prepared for all species in this study. The rachis was soaked in liquid paraffin at temperatures in the range of 60–62 °C for 2 days. The paraffin-embedded rachis sectioned sample was transferred onto a Low-E slide (Kevley Technologies, Chesterfield, OH, USA), and embedded paraffin of the rachis sample section was removed by xylene washing before SR-FTIR microspectroscopic mapping.

EDC protein secondary structure prediction

Expression profiles in feathers and protein sequences of the epidermal differentiation complex (EPC) of chickens were annotated from Lin et al. (2021a) and Lin et al. (2021b). Secondary structures of EDC proteins were predicted using JPred4 (https://www.compbio.dundee.ac.uk/jpred/, accessed on 3 December 2021) (Drozdetskiy et al., 2015).

Attenuated total reflection Fourier transform infrared spectroscopy

The attenuated total reflection Fourier transform infrared (ATR-FTIR) spectroscopy analysis system uses a modulated infrared beam to enter an infrared transparent crystal with a high refractive index in the spectral range 4,000–650 cm−1 at 45° of incidence. In this study, a high-pressure clamp is utilized to pressure the rachis medulla and dorsal cortex sample to make contact with the surface of the ATR crystal (PIKE MIRacle™, 3-reflection diamond/ZnSe crystal). Furthermore, the evanescence wave of the modulated infrared beam occurs on the contact surface of the ATR crystal; the depth of penetration (dp) of the propagating evanescence wave entering the sample ranges from approximately a hundred nanometers to several micrometers, as shown in the equation shown below; and the FTIR spectra of the feather samples were collected by using an FTIR spectrometer coupled with an LN-cooled MCT detector. dp=λ2πn1sinθ−n1/n22

where λ is the wavelength of incident mid-infrared light, θ is the incident angle of the crystal, and n 1 and n 2 are the refractive indices of the sample and ATR crystal, respectively. The FTIR spectra were acquired in the range of 4,000-650 cm−1 and accumulated 128 scans at a spectral resolution of 4 cm−1. The penetration (dp) of the infrared evanescent wave propagating into the sample was calculated to be approximately 2.15 µm at wavenumber 1,650 cm−1 (the refractive index of the feather is approximately 1.55; the refractive index of the crystal is 2.4; the incident angle is 45°; and the wavelength is approximately 6.06 micrometers for wavenumber 1,650 cm−1). Three samples of flight feathers of each avian species were analyzed.

Synchrotron-radiation-based Fourier transform infrared microspectroscopy

The spectral maps of the sectioned flight feather samples were acquired at the SR-FTIRM endstation of TLS 14A1 of the National Synchrotron Radiation Research Center (NSRRC) in Taiwan, including an FTIR spectrometer (Nicolet 6700, ThermoFisher Scientific, Madison, WI, USA) coupled with a confocal infrared microscope (Nicolet Continuum; ThermoFisher Scientific, Madison, WI, USA). The detailed procedures of SR-FTIRM mapping experiments are described in Lee et al. (2017). The confocal aperture of the confocal infrared microscope was set to 15 × 15 µm2 and 10 × 10 µm2 of step size for roast scanning of the sample area of interest. The Am I band is the main characteristic absorption of carbonyl groups of the peptide bond framework, showing lower vibration wavenumbers than those of chemicals with carbonyl functional groups due to the hyperconjugation of peptide bonds and forming various PSSs by intermolecular and intramolecular hydrogen bonds. The percentages of corresponding PSSs of feather samples in the Am I band spanning the spectral range of 1,720–1,580 cm−1 were resolved by using the Peak Resolve function in OMNIC™. The automatic atmospheric suppression function in OMNIC™ was employed to eliminate the rovibrational absorption contributions from carbon dioxide and water in the ambient air. Three samples of flight feathers of each avian species were analyzed.

Results

The flight feather rachis is parameterized along the longitudinal direction of the shaft as the z-axis (Z), which was oriented along the shaft for the length from the proximal end at the SUR (superior umbilical region, junction of the calamus, and rachis) (0.0 Z) to the distal tip (1.0 Z). The cross-section of the rachis of an avian feather of different species ranges from 0.2 Z to 0.5 Z. (Fig. 1). The Am I band spanning the spectral range of 1,720–1,580 cm−1 was deconvoluted to resolve PSSs for the am I band of the feather rachis cortex, as shown in Fig. 2. The wavenumbers of the deconvoluted PSS components were assigned to β-turns (1,614 cm−1), parallel β-sheets (1,627 cm−1), triple helix or β-sheets (1,637 cm−1), random coils (1,648 cm−1), α-helix (1,657 cm−1), β-turns (1,668 cm−1), parallel β-strand in β-sheets (1,683 cm−1), and anti-parallel β-strand in β-sheets (1,693 cm−1) (Petibois & Deleris, 2006; Petibois et al., 2006; Yang et al., 2015) (Fig. 3).

Figure 2 The representative SR-FTIR spectra of feather shaft medulla of different birds.

From top to bottom, they were domestic chickens, mallards, African sacred ibis, crested goshawks, collared owls, budgerigars, and zebra finches.

Figure 3 The spectral deconvolution of the Amide I band of the rachis medulla of domestic chicken flying feathers.

The representative spectral deconvolution of the eight resolved components spaned in the Am I absorption band. The black line was the calculated spectrum after the curve fitting process, and the red line is the normalized original Am I absorption band.

The cross-sectional flight feather rachises of seven avian species have corresponding characteristic shapes and unique anatomies (Fig. S1). The characteristic rectangular shape was observed in the cortex of the flight feather rachis for those in chicken, mallard, goshawk, budgerigar, and zebra finch. Interestingly, round or elliptical shapes of the cortex of flight feather rachises of sacred ibis and owls were identified. Furthermore, a foam-like medulla filled in the hollow space of the cortex of these flight feathers (Lingham-Soliar, 2017; Wang & Meyers, 2017; Chang et al., 2019; Deng et al., 2021). The morphological structures of the flight feather medulla differ among these species; for example, the structure in the medulla in the flight feathers of sustained flyers and birds of prey shows a V-ridge shape, and the characteristic hollow shape is observed on the dorsal part of the medulla, suggesting that the organization of the medulla is critical for providing mechanical support (Chang et al., 2019).

Spectral specificity of the cortex and medulla using attenuated total reflection Fourier transform infrared (ATR-FTIR) spectroscopy

The ratio of the absorption component band of the calculated PSS to the total area of Am I is shown in Fig. 3 (Figs. S2 & S3). In general, all medulla and cortex samples showed strong absorption attributed to PSS β-sheets, and β-keratins are well understood as major components of feathers. Additionally, the peak area percentage of approximately 10–15% attributed to characteristic absorption of the triple helix for collagen was resolved in the Am I band, consistent with the genomic expression of DNA related to collagens (Ng et al., 2015). Two characteristic absorption bands spanning the range of 1,641–1,623 cm−1 and the range of 1,695–1,670 cm−1 have been indicated as an antiparallel β-strand in a β-sheet structure for keratin (Zhang, Senak & Moore, 2011). Therefore, the peak-height ratio H1,695/H1,630 was suggested to be proportional to the relative distribution of the antiparallel β-strand component (1,695 cm−1) to the parallel β-strand component (1,630 cm−1) in a β-sheet structure (Chirgadze & Nevskaya, 1976). Previous studies of pelican and seagull feathers showed that the β-sheets primarily existed in the antiparallel conformation (Fraser & Suzuki, 1965). Collectively, the β-sheet structure was expected to be dominant in these feather samples. Although some epidermal differentiation complex (EDC) genes are expressed in feathers (Lin et al., 2021a; Lin et al., 2021b), most EDC members, especially those higly expressed in feathers, do not contain distinct secondary structures in predictions (Table S1).

The calculated peak area of the α-helix structure was approximately 9–12% in the Am I band for all feather samples; hence, we proposed that α-keratins might develop in mature flight feathers (Fig. 4). β-sheet structures were observed in the α-keratin filaments, which were suggested to increase the mechanical strength of feathers (Parry & North, 1998; Parry et al., 2002; Kreplak et al., 2004) to reduce stress during flight. Therefore, the content of α-keratins would be underestimated by the suggestion of only α-helix contribution in the structure of α-keratins. Based on our findings, the structure of β-sheets developed from α- and β-keratins plays an essential role in higher levels of assembly and in determining their mechanical properties (Fraser & Parry, 2009). The helical content was slightly higher in the flight feather cortex of chickens, ducks, and sacred ibis than in the medulla but was slightly lower in the flight feather cortex of goshawk, owl, budgerigar, and zebra finch (Table S2).

Figure 4 The ratio of the area under the curve of each PSS to the total area of Amide I band under the curve fitting of the Amide I absorption spectrum of the feather rachis cortex (A) and medulla (B).

From left to right on the horizontal axis are the absorption peaks of each secondary structure from low to high wavenumbers, and the vertical axis is the percentage of the area.

In addition, we also estimated proteins other than α- and β-keratins, in which distinct features are observed in the FTIR spectrum. The calculated contribution of the triple-helix structure in the Am I band was characteristic absorption for collagen type I; the other absorption bands were observed at 3,279 cm−1, 3,052 cm−1, 1,545 cm−1, 1,292 cm−1, and 1,260 cm−1 and assigned to amide A, amide B, Am II, and amide III (Am III) bands, respectively (Fig. 2). In the study, the calculated results showed that the cortex of the flight feather might contain 11–16% collagen, whereas the medulla contained 13–16% collagen. The collagen content was slightly higher in the medulla than in the cortex, except for budgerigar and zebra finch.

Spectral map of PSSs of feather rachis sections

We extended the investigation of PSSs of proteins packed within the medulla by using lateral-resolved synchrotron-radiation-based Fourier transform infrared microspectroscopy (SR-FTIRM). The flight feather rachis samples of each species were sectioned by a series of continuous sectioning of the same rachis position, and herein, three cross-section samples were prepared and analyzed for each species. Lateral-resolved FTIR spectral images were constructed by using the peak height of the Am I band for the feather shaft medulla of seven species of birds, as shown in Fig. 5. The absorbance of the Am I band of the medulla for these birds varied among different regions, consistent with the suggestion of the varied density of packed cells in regions (Chang et al., 2019) (Figs. S4–S10), so that corneous protein was not evenly distributed in the medulla. The relative population of PSSs, such as α-helices, β-sheets, triple-helices, and random coil secondary structures, can be resolved by using Peak Resolve software in OMNIC. The analysis of the FTIR spectra of cross-section samples showed that a higher percentage of amino acids can be expected for α-helical peptides developed within the medulla cells of the flight feathers than has been reported and is not usually observed in β-sheet peptides (Ng et al., 2012) (Fig. S8).

Figure 5 Representative spectral images of Am I band with SR-FTIRM for Feather shaft medulla of seven species of birds.

(A) The area of red grid boxes is the mapping area of the spectral image, and each red grid box (10 × 10 mm2) of sampling point is the sub-area where the absorption spectrum was measured. (B) Spectral images of cross-section feather samples were constructed by integrating the peak height of the Amide I band.

Discussion

α- and β-keratins of the medulla and cortex in different bird species

In this study, the accumulation of α- and β-keratins was observed in different parts of the rachis and ramus. α-Keratin has an important role in the early formation of rachises, barbs, and barbules, although feather- β-keratins are the major component of feathers. A deletion in two α-keratin genes (KRT75L4 and KRT6A (previously referred to as KRT75)) is associated with the frizzled feather phenotype (Ng et al., 2012; Dong et al., 2018). The rachis medulla with a defect in the KRT6A (previously known as the KRT75 gene) gene would cause the rigidity of feathers to decrease due to incomplete development and the rachis to bend (Ng et al., 2012). Despite this progress, few studies have revealed how the rachis becomes structured.

Cytoplasmic networks of intermediate filaments were built by obligate heteropolymers of types I and II α-keratins. The cornification process of claws, scales, beaks, and feathers of birds requires the deposition of β-keratins onto scaffolds of epidermal α-keratin filaments (Alibardi, 2013). Transcriptomic and evolutionary analyses were facilitated by manual annotation and curation of α- and β-keratin genes in avian genomes (Greenwold & Sawyer, 2010; Greenwold et al., 2014; Ng et al., 2014), suggesting that various combinations of α- and β-keratins contribute to morphological and structural diversity among avian epidermal appendages, with intrafeather differences largely attributed to differential expression of feather- β-keratins (Greenwold et al., 2014; Ng et al., 2014; Wu et al., 2015).

The flight feather cortex of ibis, goshawk, owl, budgerigar, and zebra finch contains higher helical content in the cortex of the flight feather than in the medulla compared to chickens and ducks who mainly live on the ground (Table S1). Goshawks and owls spend more time in woodlands and initiate mainly attacks on their prey from perches, whereas zebra finches and budgerigars are flappers with excellent maneuverability. Their flight feathers should endure stronger mechanical forces acting on the cortex in a short time. Higher contents of α-keratin may suggest that the mechanical properties provided by α-keratins in the cortex of flight feathers may have been previously disregarded. Further studies, including more avian species sampled from wide ranges of phylogenetic lineages, are required to perform a solid comparison of PSSs in birds of different flight modes.

Although it has been reported that unfavorable conditions such as nutrient-poor conditions in captivity during molting may affect the quality of avian feathers in some species of birds (Murphy, King & Lu, 1988; Dawson et al., 2000), we can eliminate this possibility because we did not observe that feathers from aviaries (chickens, ducks, budgerigar, and zebra finch) are significantly different from those derived from the wild (ibises, goshawks, and owls). Additionally, birds kept in aviaries are either domesticated birds or pet birds and should already be well adapted to artificial environments.

Collagen in the rachis

The FTIR spectra of the medulla of the feather rachis revealed crucial spectral evidence for collagen type I, as strongly supported by the characteristic absorption of the Am III band of protein in the range of 1,290–1,260 cm−1, glycan residual (oligosaccharide residue) chemically attached to collagen in the range of 1,200–900 cm−1, and triple-helix resolved at 1,637 cm−1 in Am I band (Cooper & Knutson, 1995; Belbachir et al., 2009; Lee et al., 2017). Based on these findings, we strongly suggested that collagen type I develops in the medulla of the feather rachis.

Collagen genes are differentially expressed in regenerating feathers of different morphotypes (Ng et al., 2015), and collagens might play morphogenetic roles during embryonic feather development (Mauger et al., 1982). Type VI collagen is regulated by mesenchymal cell behaviors during early feather development in embryonic chickens (Kobayashi et al., 2005). The distribution of collagen types I, III, and V becomes heterogeneous during the formation of placode and disappears from the apex of growing feather buds, while their collagen fiber becomes denser in the rudiment (Sengel, 1990). Collagens have been proposed to have critical roles in the development of feather rudiments (Goetinck & Sekellick, 1972). However, the role of collagen in mature feathers needs further investigation, so it is interesting to determine if collagens play a mechanistic role in feathers.

Feather rachis morphology

The medulla of the rachis is composed of keratinocytes that are developed to grow into hollow, spongy, and porous structures (Sullivan et al., 2019). The types of keratinocytes also develop into different growth directions, pore sizes, and pore shapes; furthermore, different pore sizes and medullary arrangements are well understood in the feather shaft medulla for birds with different flight abilities and flight modes (Chang et al., 2019). The ratio of the diameter of the average pore size to the weight in the medulla of the sparrows that flutter quickly is relatively high, while the elongation and orientation angles of aligned cell bands in the medulla of the soaring flight differ from those of other species.

Based on our findings, a greater area of the hollow cell depletion zone was observed in the medulla of the cross-section feather shaft, which may enable the feathers to resist greater mechanical stresses during flight, especially for birds with a better flight ability (Wang & Meyers, 2017; Chang et al., 2019). In addition, the cortex of the shaft of sustained flyers also tends to be thicker on dorsal and ventral sides and thinner on lateral sides (Wang & Meyers, 2017; Wang & Sullivan, 2017; Chang et al., 2019). Accordingly, this may represent the difference in the medulla or cortical structure of the feather shafts of birds with various flight modes, giving them different flight abilities.

Conclusions

In this study, ATR-FTIR spectroscopy and SR-FTIRM were applied to resolve the components of PSSs and to provide spectral images for cross-sectioned feather samples with the two-dimensional distribution of the Am I band in flight feathers of several avian species. We observed the ratio of α-keratins/β-keratins and collagens in both the cortex and medulla of flight feathers based on the information extracted from the Am I bands. A significant amount of α-keratins and collagens has been detected in flight feather shafts, and these proteins vary in birds with different flight abilities, suggesting that they may contribute to different mechanical properties. Both SR-FTIRM and ATR-FTIR are nondestructive methods and provide powerful methods for understanding the content and distribution of these structural proteins in mature feathers.

Supplemental Information

Table S1 EDC protein secondary structure prediction

Expression profiles in feathers and protein sequences of the epidermal differentiation complex (EPC) of chickens were annotated from Lin et al. (2021a); Lin et al. (2021b). Secondary structures of EDC proteins were predicted using JPred4 (https://www.compbio.dundee.ac.uk/jpred/, accessed on 3 December 2021) (Drozdetskiy et al., 2015).

Click here for additional data file.

Table S2 Secondary structure calculation from flight feather rachis FT-IR spectra

Click here for additional data file.

Figure S1 Cross-sections of the cortex of flight feather rachis for seven species of birds

Click here for additional data file.

Figure S2 Curve fitting of the Amide I bands of the feather rachis medulla for seven species of birds

The black line is the fitted spectrum, and the red line is the normalized original spectrum.

Click here for additional data file.

Figure S3 Curve fitting of the Amide I absorption bands of the feather shaft cortex of seven species of birds

The black line is the fitted spectrum, and the red line is the normalized original spectrum.

Click here for additional data file.

Figure S4 SR-FTIR spectral image of β-turns (1,614 cm −1) ratio in Amide I band

Click here for additional data file.

Figure S5 SR-FTIR spectral image of parallel β-strand (1,627 cm −1) in Amide I band

Click here for additional data file.

Figure S6 SR-FTIR spectral image of Triple-helix (1,637 cm −1) in Amide I band

Click here for additional data file.

Figure S7 SR-FTIR spectral image of Unordered structure (1,648 cm −1) in Amide I band

Click here for additional data file.

Figure S8 SR-FTIR image of α-helix (1,657 cm −1) in Amide I band

Click here for additional data file.

Figure S9 SR-FTIR spectral image of parallel β-strand (1,683 cm −1) in Amide I band

Click here for additional data file.

Figure S10 SR-FTIR spectral image of Anti-parallel β-strand (1,693 cm −1) in Amide I band

Click here for additional data file.

Data S1 Amide I absorption band of feather rachis cortex of seven species of birds

Each data point is the Amide I absorption value and fitting curve under nondestructive attenuated total reflection Fourier transform infrared spectroscopy (ATR-FTIR). The data of each individual flight feather were shown in different worksheets.

Click here for additional data file.

Data S2 Amide I absorption band of feather rachis medulla of seven species of birds

Each data point is the Amide I absorption value and fitting curve under nondestructive attenuated total reflection Fourier transform infrared spectroscopy (ATR-FTIR). The data of each individual flight feather were shown in different worksheets.

Click here for additional data file.

We thank Profs. Chih-Feng Chen and Si-Min Lin, the Raptor Research Group of Taiwan, and Taipei Zoo for providing us with essential feather samples. Profs. Wen-Tau Juan and Po-Yu Chen provided excellent suggestions.

Additional Information and Declarations

Competing Interests

Author Contributions

Animal Ethics

Data Availability

The authors declare there are no competing interests.

Pin-Yen Lin conceived and designed the experiments, performed the experiments, analyzed the data, prepared figures and/or tables, and approved the final draft.

Pei-Yu Huang performed the experiments, prepared figures and/or tables, and approved the final draft.

Yao-Chang Lee and Chen Siang Ng conceived and designed the experiments, prepared figures and/or tables, authored or reviewed drafts of the paper, and approved the final draft.

The following information was supplied relating to ethical approvals (i.e., approving body and any reference numbers):

The use of feathers from wild animals for research was approved by the Forestry Bureau of the Council of Agriculture of Taiwan (R.O.C.) under case no. 1091617349 (4/22/2020).

The following information was supplied regarding data availability:

The raw data is available in the Supplemental Files.

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
