# Peer review of "Analysis and comparison of protein secondary structures in the rachis of avian flight feathers"

_PeerJ, doi:10.7717/peerj.12919_

## Round 0.1 · original submission · Major Revisions

As you can see, reviewers raised multiple serious concerns and indicated that major revision is needed. Please address all critiques and amend your manuscript accordingly.

Reviewer 1 ·

Basic reporting

The aim of this study was to determine protein secondary structures (PSS) of flight feathers of seven bird species. Although potentially interesting, there are several points that preclude recommendation of acceptance.

The use of the English language needs to be improved. There are also typos. For example, in the abstract: he.

Experimental design

The research question is not well defined.

The data analysis is not convincing. Which standards were used for PSS?

The curve fitting in Figure 3 appears very questionable. A detailed description and validation of the approach is necessary. Statistical analysis is required.

Legend of Figure 5: Was hair investigated?

Validity of the findings

The validity of the findings is not demonstrated by the authors.
The data analysis is not clear.
Statstical anaylsis is missing.
Conclusions are not well stated.

Reviewer 2 ·

Basic reporting

The manuscript entitled “Analysis and comparison of protein secondary structures in the rachis of avian flight feathers” is a timely publication at the height of crystallography. Furthermore, the comparison of the species is well thought out and provides a encompassing view of flight feather’s secondary structure. However, I have several general comments and one major revision that should be addressed before publication.
Major revision:
1) The authors attribute the identified secondary structures to only alpha-keratins, beta-keratins and collagen. However, we now know that proteins comprising the EDC genetic locus are widely expressed in feathers (Strasser et al. 2014 and more recent papers). The authors need to justify their attribution of these structures to just three types of proteins. Secondary structures of proteins are easily calculated using online bioinformatics tools. What are the secondary structures of these other EDC proteins? Could any of the beta-sheets or alpha-helixes be attributed to those proteins? Beta-sheets and alpha-helixes are very common among proteins performing a myriad of functions. Therefore, I think the authors should calculated the secondary structures of the EDC genes identified in the chicken (Strasser et al., 2014) in Results and justify their conclusion of those structures only being attributed to three protein types in the Discussion.
Minor revisions:
1) The manuscript should be carefully read over by a native English speaker. There are several grammar issues and/or awkward sentences throughout the ms. For example these sentences should be revised (these are only some examples): Lines 31-32, 56-58, 89-90.
2) The introduction should include specific information on the flight modes of these birds. Also and related, the information in the first three paragraphs of the Discussion and first paragraph of the “Collagen in the rachis” of the Discussion should have been in the Introduction.
3) The information in the paragraph found on lines 201-210 should have references as this is not novel information.
4) The first figures cited in the text (lines 193-195) are Figure 1 and Figure 3. Therefore, Figure 3 should be renamed Figure 2.

Experimental design

no comment

Validity of the findings

no comment

Additional comments

no comment

Reviewer 3 ·

Basic reporting

The language: the English language should be improved to ensure that an international audience can clearly understand your text. Some examples where the language could be improved include for example lines 31-34, 40-42, 53-56, 76-78, 80-81, 89-90 (the adaption of the rachis medulla for flighting), and many other throughout the ms, where the current phrasing makes comprehension difficult, some sentences are ungrammatical or simply constitute sentence fragments. I would like to suggest you ask a colleague who is proficient in English and familiar with the subject matter review your manuscript, or contact a professional editing service.
The Introduction is well written and adequately introduces the research questions.
The selection of literature is good and relevant. See section 4 “Additional comments” for minor remarks on the subject.
In my opinion, the structure of the manuscript meets PeerJ standards and disciplinary norms.
I thank you for supplying the raw data, which is accordance with the PeerJ standards. However, your supplemental Excel sheets require a detailed description of the variables contained therein, e.g. in the form of a legend, to be useful to future readers and users.

Experimental design

The aim of this study was to analyze the molecular structure (protein secondary structure, or PSS) of flight feathers in seven bird species, using up-to-date spectroscopy methods (ATR-FTIR and SR_FTIRM). Then the Authors sought to establish whether there was a relationship between the flight mode used by the species studied and the secondary protein structure composition of their feathers. Previous studies showed that mechanical properties of avian flight feathers (e.g. stiffness) depend mainly on the rachis structure. The Authors rightly suggested that the keratin and corneous protein composition is also important here. The research questions are well defined, relevant and meaningful.
To the best of my knowledge, the methods you used to molecularly analyze the feather composition are appropriate and up-to-date. These methods are described with sufficient detail and information to replicate.
My reservations about the methods concern the selection of feathers for analysis. You wrote that you analyzed “flight feathers”. But which feather? Rectrices, remiges, primaries or secondaries? Please specify.
Another problem is the choice of bird species – you sampled feathers from aviary birds (Mallards and chicken) and from wild birds (ibises). You did not account for the possibility that unfavorable conditions during molt in captivity may cause birds to deposit low quality keratin in growing feathers (e.g. Murphy et al. Can J Zool 66:1403-1413, 1998; Dawson et al. Proc R Soc Lond B: 267: 2093-2098, 2000). This could affect the results of your analysis. I warmly suggest you to address this issue in the paper.
You have classified the Mallard as a weak flier. This is not a good classification. Mallards are strong fliers, performing daily migrations between foraging and resting sites, migrations to molt sites, and even long-distance seasonal migrations. They have short, but high aspect ratio and high wing loading wings. Such wings produce high induced power and are adapted for fast and sustained flight over long distances (e.g. Rayner, Form and function of avian flight, Current Ornithology, 1988). Wing flight feathers in Mallards are certainly structurally adapted to withstand high aerodynamic forces.
And last but not least. You aimed at looking for a relationship between feather structure and flight mode in study birds. But you did not perform such an analysis. I would suggest you to use statistics to compare flight feather morphologies between strong and weak flying species in your sample.

Validity of the findings

The results obtained in this study are undoubtedly worth publishing. The rationale and benefits of peer-reviewed research to the scientific discipline are clearly stated and justified.
Feather molecular structure analysis data are well presented. They are robust. However, I see no quantitative or statistical justification for the claims (lines 362-369 and 378-380) that differences in feather protein composition were found between good and poor flyers.
Conclusions are well stated in terms of the molecular structure of the feathers. However, I see no statistical justification for the statement that good and poor flying birds differ in the protein composition of their flight feathers. I strongly encourage you to provide such a justification.

Additional comments

Citations:
Lines 60-61: in this case, the following papers should be cited: Feduccia & Tordoff Science 203:1021–1022, 1979; Norberg, Vertebrate flight, Springer, 1990.
Reference list:
The names of the journals are given usually in abbreviated form, but sometimes in full (e.g. lines 414-415). Please standardize.
Bonser, R. and P. Purslow (1995). "The Young's modulus of feather keratin." J Exp Biol 198(Pt420 4): 1029-1033 – this citation contains an error – what is “PT410” that is shown in parentheses?
Please check all citations and make them uniform.

---

## Round 0.2 · accepted · Accept

All issues pointed out by the reviewers were addressed, and the revised manuscript is acceptable now.

Reviewer 1 ·

Basic reporting

The quality of the text has been improved.

Experimental design

no comment

Validity of the findings

The reporting of the results has been improved.

Reviewer 2 ·

Basic reporting

I think the authors adequately addressed all of my initial concerns. Therefore, I think the ms is ready for publication.

Experimental design

I think the authors adequately addressed all of my initial concerns. Therefore, I think the ms is ready for publication.

Validity of the findings

I think the authors adequately addressed all of my initial concerns. Therefore, I think the ms is ready for publication.